# Predictive factors of clinical outcomes in patients with COVID-19 treated with tocilizumab: A monocentric retrospective analysis

Giulia Cassone[1,2], Giovanni Dolci[3]*, Giulia Besutti[2,4], Luca Braglia[5], Paolo Pavone[6], Romina Corsini[6], Fabio Sampaolesi[6], Valentina Iotti[4], Elisabetta Teopompi[7], Marco Massari[6], Matteo Fontana[8], Giulia Ghidoni[8], Anaflorina Matei[9], Stefania Croci[10], Emanuele Alberto Negri[11], Massimo Costantini[5], Nicola Facciolongo[8], Carlo Salvarani[1,12]

1 Rheumatology Unit, IRCCS Arcispedale Santa Maria Nuova, Azienda Unità Sanitaria Locale-IRCCS di Reggio Emilia, Reggio Emilia, Italy, 2 Clinical and Experimental Medicine PhD Program, University of Modena and Reggio Emilia, Modena, Italy, 3 Infectious Disease Unit, University of Modena and Reggio Emilia, Modena, Italy, 4 Radiology Unit, Department of Imaging and Laboratory Medicine, Azienda USL-IRCCS di Reggio Emilia, Reggio Emilia, Italy, 5 Azienda USL-IRCCS di Reggio Emilia, Reggio Emilia, Italy, 6 Infectious Disease Unit, Azienda USL-IRCCS di Reggio Emilia, Reggio Emilia, Italy, 7 SOC Internistica Multidisciplinare, Ospedale Civile Guastalla, Azienda USL-IRCCS di Reggio Emilia, Reggio Emilia, Italy, 8 Pneumology Unit, Azienda USL-IRCCS di Reggio Emilia, Reggio Emilia, Italy, 9 Department of Anesthesia and Intensive Care, Azienda USL-IRCCS di Reggio Emilia, Reggio Emilia, Italy, 10 Clinical Immunology, Allergy and Advanced Biotechnologies Unit, Azienda USL-IRCCS di Reggio Emilia, Reggio Emilia, Italy, 11 High Intensity Unit, Azienda USL-IRCCS di Reggio Emilia, Reggio Emilia, Italy, 12 Rheumatology Unit, University of Modena and Reggio Emilia, Modena, Italy

* giodolci@hotmail.it

**Data Availability Statement:** Data cannot be shared publicly because of local privacy policies. Data are available from the Azienda USL-IRCCS Santa Mari Nuova di Reggio Emilia (contact via

## Abstract

### Objective

The aim of this retrospective observational study is to analyse clinical, serological and radiological predictors of outcome in patients with COVID-19 pneumonia treated with tocilizumab, providing clinical guidance to its use in real-life.

### Method

This is a retrospective, monocentric observational cohort study. All consecutive patients hospitalized between February the 11th and April 14th 2020 for severe COVID-19 pneumonia at Reggio Emilia AUSL and treated with tocilizumab were enrolled. The patient's clinical status was recorded every day using the WHO ordinal scale for clinical improvement. Response to treatment was defined as an improvement of one point (from the status at the beginning of tocilizumab treatment) during the follow-up on this scale. Bivariate association of main patients' characteristics with outcomes was explored by descriptive statistics and Fisher or Kruskal Wallis tests (respectively for qualitative or quantitative variables). Each clinically significant predictor was checked by a loglikelihood ratio test (in univariate logistic models for each of the considered outcomes) against the null model.

presidio.ospedaliero@ausl.re.it) for researchers who meet the criteria for access to confidential data.

**Funding:** The authors received no specific funding for this work.

**Competing interests:** The authors have declared that no competing interests exist.

## Results

A total of 173 patients were included. Only hypertension, the use of angiotensin-converting enzyme inhibitors, PaO$_2$/FiO$_2$, respiratory rate and C-reactive protein were selected for the multivariate analysis. In the multivariable model, none of them was significantly associated with response.

## Conclusions

Evaluating a large number of clinical variables, our study did not find new predictors of outcome in COVID19 patients treated with tocilizumab. Further studies are needed to investigate the use of tocilizumab in COVID-19 and to better identify clinical phenotypes which could benefit from this treatment.

## Introduction

An aberrant immune-response characterized by uncontrolled hyperinflammation has been described in severe COVID-19 patients, and elevated blood levels of interleukin 6 (IL-6) have been related to a fatal outcome [1]. Tocilizumab (TCZ) is a humanized monoclonal antibody against IL6 receptor approved for the treatment of autoimmune rheumatic diseases. It has been administered as an off-label drug in the treatment of severe manifestation of COVID-19 with promising results [2, 3] but did not reduce mortality in the first published randomized controlled trials [4–7]. Nevertheless, recent data from two randomized platform trials, REMAP-CAP [8] and RECOVERY [9], showed a significant improvement in survival in patients with advanced-stage COVID-19 that underwent TCZ therapy.

Thus, the role of tocilizumab in COVID-19 treatment appears to be beneficial in selected subgroups of patients [10]. Early identification of personalized and efficacious treatment options for COVID-19 pneumonia, according to patient-specific or disease-specific parameters, could lead to a beneficial effect on prognosis and long-term outcome.

The aim of this retrospective observational study is to analyse clinical, serological and radiological predictors of outcome in patients with COVID-19 pneumonia treated with tocilizumab, providing clinical guidance to its use in real-life.

## Materials and methods

### Patients and case definition

This is a retrospective, monocentric observational cohort study performed at Reggio Emilia AUSL, at two different sites, the central research hospital of Reggio Emilia and Guastalla Hospital (Reggio Emilia province, Italy). We enrolled all consecutive patients hospitalized between February the 11[th] and April 14[th] 2020 for severe COVID-19 pneumonia and treated with TCZ.

The study was approved by *Comitato Etico* Area Vasta Emilia Nord. For most patients it was not possible to obtain a written consent as in the first phase of the pandemic at our site we did not want anything touched by the patients to be kept in the patients' records. Nevertheless, verbal consent was always obtained, recorded, and signed by the attending physicians. This consent procedure was approved by both national and local ethics committee.

SARS-CoV2 infection was diagnosed at Hospital admission by a positive reverse-transcriptase polymerase chain-reaction in a respiratory tract specimen. COVID-19 pneumonia was

confirmed if chest X-rays and/or high-resolution computed tomography (HRCT) scan showed suggestive findings [11–13].

## Treatment

Tocilizumab was administered by intravenous (iv) or subcutaneous (sc) formulations. Sc TCZ was used in some patients because iv tocilizumab was not available for a period of time.

Iv TCZ was prescribed 8 mg/kg (maximum dose per single infusion: 800 mg), first dose at time 0 and a second dose after 12 hours. Sc TCZ was administered as 162 mg vials for a total of 2 to 4 administration, depending on patient's weight.

Suggested clinical features for TCZ-therapy eligibility were the evidence of a severe pneumonia (oxygen saturation at rest on room air ≤93% and/or arterial oxygen partial pressure (PaO2)/oxygen concentration (FiO2) ≤300 mmHg), the presence of exaggerated inflammatory response (body temperature> 38°C; serum C reactive protein greater than or equal to 10 mg/dl or at least double the basal value) and absence of contraindications to TCZ therapy.

Together with TCZ therapy, patients could undergo other pharmacological treatments according to the standard of care that varied during the time-span considered for the study [14–16], namely: oxygen supply, hydroxychloroquine, lopinavir/ritonavir or darunavir/cobicistat, enoxaparin or unfractionated calcium heparin, glucocorticoids.

## Data collection

Data were collected from both paper and electronic clinical records.

A standardized protocol with predefined laboratory tests at admission and during the follow-up was followed for all hospitalized COVID-19 patients from March 31st. Moreover, from both paper and electronic clinical records we collected information about hospital discharge, the condition of the patients at hospital discharge, the type of respiratory support and death. Patients' past medical history, including comorbidities and current medications at home as well as glucocorticoids use before or after TCZ administration during the follow-up time was also recorded.

## Radiological data

CT scans were performed using one of three scanners (128-slice Somatom Definition Edge, Siemens Healthineers care; 64-slice Ingenuity, Philips Healthcare; 16-slice GE Brightspeed, GE Healthcare Medical System) without contrast media injection, with the patient in supine position, during end-inspiration. Scanning parameters were: tube voltage 120 KV, automatic tube current modulation, collimation width 0.625 or 1.25 mm, acquisition slice thickness 2.5 mm, and interval 1.25 mm. Images were reconstructed with a high-resolution algorithm at slice thickness 1.0/1.25 mm.

Computerized tomography (CT) scans performed at emergency department presentation were retrospectively reviewed by a single radiologist with 15 years of experience, collecting the presence/absence of ground-glass opacities, consolidations, crazy-paving pattern and reversed halo sign, as well as the extension of pulmonary lesions using a visual scoring system (< 20%, 20–40%, 40–60%, and > 60% of parenchymal involvement).

## Outcome measures

Patients were assessed daily during the hospitalization. The patient's clinical status was recorded every day using a six-category ordinal scale defined as follows: 1) not hospitalized; 2) hospitalized, not requiring supplemental oxygen; 3) hospitalized, requiring any supplemental

oxygen; 4) hospitalized, requiring non-invasive ventilation or use of high-flow oxygen devices; 5) hospitalized, receiving invasive mechanical ventilation or ECMO; 6) death. Response to treatment was defined as an improvement of one point (from the status at the beginning of TCZ treatment) during the follow-up on a six-category ordinal scale or live discharge from the hospital, whichever came first. On the contrary, a non-response to treatment was defined as at least one-point worsening of the clinical status after TCZ therapy. Patients who initially declined and afterward improved on the six-category ordinal scale were defined as non-responders. Beside response to treatment, a composite outcome of death or intubation during follow-up was considered. Patients already intubated at the moment of TCZ administration were not included for analyses regarding this outcome.

## Statistical analysis

Bivariate association of main patients' characteristics with outcomes was explored by descriptive statistics and Fisher or Kruskal Wallis tests (respectively for qualitative or quantitative variables).

Each clinically significant predictor was checked by a loglikelihood ratio test (in univariate logistic models for each of the considered outcomes) against the null model. Multivariable models were estimated for the two considered outcomes including all the variables with a p-value was less than 0.2 in the univariate models. Odds ratio (OR) (with respective 95% confidence intervals and Wald tests) were provided for univariate and multivariate logistic models. For reporting completeness, we estimated final models' area under the Receiver Operating Characteristic curve (AUC) with respective 95% confidence interval.

Statistical analysis was performed using R 3.5.2 R Core Team (2020).

## Results

A total of 173 patients were recruited for the study. For 144 patients data were available for each covariate analysed (32 females and 112 males; median age 63.98 years). Demographic, clinical, serological and radiological features of patients are summarized in Table 1. Median Charlson's score was 2 (range 0–8, IQR 2–3).

The median follow-up was 318 (IQR 51–428.5). Three patients had history of stroke or transient ischemic attack. Five patients had chronic kidney disease and one was a kidney transplant recipient. One patient had cirrhosis. Only two patients were on immunosuppressive therapy at baseline, one for ulcerative rectum-colitis and one for polymyalgia rheumatica. One patient had HIV infection.

Most patients responded to TCZ therapy (105 patients; 72.9%), 19 patients died while a total of 25 patients died or were intubated during the follow-up. Three patients received TCZ therapy during their stay in intensive care unit with oro-tracheal intubation and invasive ventilation.

TCZ was administered sc in 76 patients (52.8%) and iv in 68 (47.2%); 32 patients (22.2%) were treated with glucocorticoids before TCZ, while 61 (42.6%) received GCs together with or after TCZ treatment.

Table 2 shows univariate and multivariate models for non-response to TCZ therapy.

Hypertension, the use of angiotensin-converting enzyme inhibitors (ACE-I) at home, $PaO_2/FiO_2$, respiratory rate and C-reactive protein (CRP) were selected for the multivariate analysis. In the multivariable model, none of them was significantly associated with response, however some borderline associations may be cautiously described. Particularly, the use of ACE-I (OR = 2.52, 95%CI = 1.25–6.41, p = 0.012) and increasing CRP values (OR for unit increase = 1.66, 95%CI = 1.08–3.28, p = 0.05) were inversely associated with response to TCZ

**Table 1. Demographic, clinical, serological and radiological features of COVID19 patients treated with tocilizumab at baseline.**

| | WHOLE POPULATION | NON-RESPONDERS | RESPONDERS | p | DEATH/ INTUBATION | ALIVE/NOT INT. | p |
|---|---|---|---|---|---|---|---|
| | (n = 144) | (n = 39) | (n = 105) | | (n = 25) | (n = 119) | |
| | N (%) or median (IQR) | N (%) or median (IQR) | N (%) or median (IQR) | | N (%) or median (IQR) | N (%) or median (IQR) | |
| Age | 63.98 (56.99–71.34) | 65.13 (58.21–69.62) | 63.16 (57.02–71.99) | 0.660 | 67.39 (62.46–70.08) | 62.89 (55.01–71.53) | 0.070 |
| Female/Male ratio | 32/112 (22.22%/ 77.78%) | 10/29 (25.64%/ 74.36%) | 22/83 (20.95%/ 79.05%) | 0.650 | 7/18 (28%/72%) | 25/94 (21.01%/ 78.99%) | 0.440 |
| Follow-up (days) | 12 (8–19.25) | | | | | | |
| Days from symptoms onset to admission | 7 (5–10) | 6 (4–9.5) | 7 (5–10) | 0.210 | 6 (3–8) | 7 (5.5–10) | 0.038 |
| Days from admission to tocilizumab administration | 2 (1–5) | 2 (1–5.5) | 2 (1–4) | 0.993 | 4 (1–6) | 2 (1–4) | 0.131 |
| Current smokers | 11 (7.64%) | 3 (7.69%) | 8 (7.62%) | 0.240 | 3 (12%) | 8 (6.72%) | 0.010 |
| Former smokers | 10 (6.94%) | 5 (12.82%) | 5 (4.76%) | | 5 (20%) | 5 (4.2%) | |
| BMI | 28.73 (26.12–32.34) | 28.53 (25.79–31.44) | 28.73 (26.12–32.58) | 0.800 | 29.3 (26.23–38.06) | 28.02 (26.12–31.99) | 0.510 |
| Hypertension | 93 (64.58%) | 30 (76.92%) | 63 (60%) | 0.077 | 20 (80%) | 73 (61.34%) | 0.110 |
| Ischemic heart disease | 10 (6.94%) | 3 (7.69%) | 7 (6.67%) | 0.990 | 2 (8%) | 8 (6.72%) | 0.690 |
| COPD | 7 (4.9%) | 3 (7.69%) | 4 (3.85%) | 0.390 | 3 (12%) | 4 (3.39%) | 0.100 |
| Diabetes mellitus | 34 (23.61%) | 12 (30.77%) | 22 (20.95%) | 0.270 | 8 (32%) | 26 (21.85%) | 0.300 |
| PaO2/FiO2 (mmHg) | 150.5 (110–231.25) | 130 (100–165) | 164 (119–240) | 0.010 | 110 (92–140) | 161 (121.5–239.5) | <0.001 |
| SO2 (%) | 94 (91.5–96) | 93 (91.4–94) | 94.1 (92–96) | 0.120 | 94 (92–94.3) | 94 (91.4–96) | 0.640 |
| Respiratory rate (acts/min) | 24 (20–28) | 26 (20–32) | 24 (20–28) | 0.340 | 28 (22–33) | 24 (20–28) | 0.070 |
| Room air at TCZ start | 10 (6.94%) | 6 (15.38%) | 4 (3.81%) | 0.020 | 2 (8%) | 8 (6.72%) | 0.010 |
| Venturi mask at TCZ start | 78 (54.17%) | 16 (41.03%) | 62 (59.05%) | | 7 (28%) | 71 (59.66%) | |
| NIV at TCZ start | 53 (36.81%) | 15 (38.46%) | 38 (36.19%) | | 14 (56%) | 39 (32.77%) | |
| OTI at TCZ start | 3 (2.08%) | 2 (5.13%) | 1 (0.95%) | | 2 (8%) | 1 (0.84%) | |
| CRP (mg/dl) | 15.03 (9.04–20.18) | 17.68 (11.57–23.06) | 14.28 (8.2–19.68) | 0.060 | 18.06 (11.93–23.05) | 14.5 (8.98–19.69) | 0.210 |
| Procalcitonin (microgr/dl) | 0.18 (0.12–0.33) | 0.22 (0.14–0.42) | 0.16 (0.11–0.33) | 0.220 | 0.17 (0.12–0.34) | 0.18 (0.11–0.33) | 0.870 |
| IL6 (pg/ml) | 60.15 (31.25–109.5) | 81.65 (41.10–171.90) | 56.75 (29.98–82.85) | 0.130 | 61.65 (36.57–115.23) | 60.15 (31.13–109.05) | 0.650 |
| LDH (U/L) | 584 (432–762) | 591 (405–810) | 569.5 (436–745.5) | 0.380 | 591 (456–820) | 569.5 (429.5–751) | 0.230 |
| Ferritin (ng/ml) | 1096 (581–2624.5) | 1394 (826–2704) | 1078 (560.5–2481.5) | 0.450 | 1560.15 (773.5–2245) | 1083 (581–2636.25) | 0.780 |
| D-dimer (ng/ml) | 812.5 (505.75–1946.25) | 1446 (543–4004.5) | 724 (463–1620) | 0.220 | 3352.5 (1776.75–5457.75) | 713 (448.25–1610.75) | 0.002 |
| Neutrophils (10³/ml) | 5475 (4145–7410) | 5780 (4500–8320) | 5230 (3945–7045) | 0.130 | 5480 (4495–9110) | 5310 (4010–7170) | 0.170 |
| Platelets (10³/ml) | 225.5 (167.75–298.75) | 203 (163–285) | 235 (178–312) | 0.110 | 204 (157–2858.5) | 227 (173–307) | 0.350 |
| Troponin | 8.5 (4.45–18.65) | 8.6 (5.15–22.55) | 8.15 (4.25–17.55) | 0.530 | 9.4 (6.25–39.43) | 8 (4.3–17.4) | 0.170 |
| Consolidations | 102 (70.83%) | 28 (71.79%) | 74 (70.48%) | 0.990 | 20 (80%) | 82 (68.91%) | 0.340 |
| Lung involvement <20% | 31 (21.53%) | 8 (20.51%) | 23 (21.9%) | 0.870 | 5 (20%) | 26 (21.85%) | 0.250 |
| Lung involvement 20–40% | 44 (30.56%) | 11 (28.21%) | 33 (31.43%) | | 4 (16%) | 40 (33.61%) | |
| Lung involvement 40–60% | 37 (25.69%) | 12 (30.77%) | 25 (23.81%) | | 8 (32%) | 29 (24.37%) | |
| Lung involvement >60% | 32 (22.22%) | 8 (20.51%) | 24 (22.86%) | | 8 (32%) | 24 (20.17%) | |
| Treatments: | | | | | | | |

*(Continued)*

**Table 1.** (Continued)

| | WHOLE POPULATION | NON-RESPONDERS | RESPONDERS | p | DEATH/ INTUBATION | ALIVE/NOT INT. | p |
|---|---|---|---|---|---|---|---|
| | (n = 144) | (n = 39) | (n = 105) | | (n = 25) | (n = 119) | |
| | N (%) or median (IQR) | N (%) or median (IQR) | N (%) or median (IQR) | | N (%) or median (IQR) | N (%) or median (IQR) | |
| **Previous (at home)** | | | | | | | |
| ACE-I | 34 (23.61%) | 15 (38.46%) | 19 (18.1%) | 0.020 | 10 (40%) | 24 (20.17%) | 0.040 |
| ARB | 29 (20.14%) | 8 (20.51%) | 21 (20%) | 0.990 | 5 (20%) | 24 (20.17%) | 0.990 |
| **During hospitalization** | | | | | | | |
| TCZ iv | 68 (47.22%) | 19 (48.72%) | 49 (46.67%) | 0.850 | 12 (48%) | 56 (47.06%) | 0.990 |
| TCZ sc | 76 (52.78%) | 20 (51.28%) | 56 (53.33%) | | 13 (52%) | 63 (52.94%) | |
| Oxygen supply | 142 (98.61%) | 39 (100%) | 103 (98.1%) | 0.990 | 25 (100%) | 117 (98.32%) | 0.990 |
| GCs before TCZ | 32 (22.22%) | 8 (20.51%) | 24 (22.86%) | 0.830 | 6 (24%) | 26 (21.85%) | 0.800 |
| GCs along with or after TCZ | 61 (42.66%) | 16 (41.02%) | 45 (42.86%) | 0.990 | 12 (48%) | 49 (41.18%) | 0.500 |
| Discharged | 124 (86.11%) | 19 (48.72%) | 105 (100%) | <0.001 | 5 (20%) | 119 (100%) | <0.001 |
| In room air | 1 (0.69%) | 1 (2.56%) | 0 (0%) | | 1 (4%) | 0 (0%) | |
| Deceased | 19 (13.19%) | 19 (48.72%) | 0 (0%) | | 19 (76%) | 0 (0%) | |

BMI: Body Mass Index; COPD: Chronic Obstructive Pulmonary Disease; PaO2/FiO2: partial arterial oxygen pressure to inspired oxygen fraction ratio; SO2: peripheral oxygen saturation; NIV: non invasive ventilation; OTI: oro-tracheal intubation; TCZ: tocilizumab; CRP: C-reactive protein; IL6: interleukin-6; LDH: lactate de-hydrogenasis; ACE-I: angiotensin converting-enzyme inhibitors; ARB: angiotensin II receptor blockers; GCs: glucocorticoids. We used Fisher tests to compare percentages and Kruskal Wallis tests to compare distribution of quantitative ones.

**Table 2. Clinical, serological and radiological features associated with NON response to tocilizumab therapy.**

| | Univariate | | | Multivariate | | |
|---|---|---|---|---|---|---|
| | OR | CI (95%) | p-value | OR | CI (95%) | p-value |
| Gender | 1.301 | 0.534–3.021 | 0.548 | | | |
| Age | 1.003 | 0.971–1.038 | 0.845 | | | |
| Time symptoms to TCZ therapy | 1.001 | 0.924–1.078 | 0.977 | | | |
| NIV or intubation at TCZ therapy | 1.308 | 0.615–2.757 | 0.481 | | | |
| Hypertension | 2.222 | 0.988–5.4 | 0.063 | 1.649 | 0.641–4.494 | 0.309 |
| Diabetes mellitus | 1.677 | 0.72–3.802 | 0.220 | | | |
| Ischemic heart disease | 1.167 | 0.242–4.447 | 0.830 | | | |
| Current smoker | 1.113 | 0.233–4.122 | 0.880 | | | |
| Former smoker | 2.968 | 0.778–11.342 | 0.102 | | | |
| ACE-I | 2.829 | 1.247–6.412 | 0.012 | 2.525 | 0.9996–6.464 | 0.0504 |
| Angiotensin receptor blockers | 1.032 | 0.395–2.501 | 0.946 | | | |
| PaO2/FiO2 | 0.993 | 0.987–0.998 | 0.013 | 0.995 | 0.988–1.0004 | 0.081 |
| Respiratory rate | 1.049 | 0.993–1.111 | 0.088 | 1.03 | 0.972–1.095 | 0.335 |
| CRP | 1.735 | 1.081–3.282 | 0.056 | 1.655 | 1.049–3.134 | 0.065 |
| Lung involvement 20–40% | 0.958 | 0.335–2.824 | 0.937 | | | |
| Lung involvement 40–60% | 1.380 | 0.483–4.096 | 0.551 | | | |
| Lung involvement >60% | 0.958 | 0.304–3.02 | 0.941 | | | |
| GCs before TCZ | 0.871 | 0.336–2.08 | 0.764 | | | |
| GCs along with or after TCZ | 0.928 | 0.435–1.947 | 0.843 | | | |
| TCZ: IV | 0.928 | 0.435–1.947 | 0.843 | | | |

**Table 3. Clinical, serological and radiological features associated with death or intubation.**

| | Univariate | | | Multivariate | | |
|---|---|---|---|---|---|---|
| | OR | 95% CI | p-value | OR | 95% CI | p-value |
| Gender | 1.382 | 0.459–3.738 | 0.539 | | | |
| Age | 1.047 | 1.002–1.098 | 0.049 | 1.034 | 0.978–1.101 | 0.258 |
| Time symptoms to TCZ therapy | 0.949 | 0.846–1.057 | 0.359 | | | |
| NIV or intubation at TCZ therapy | 3.151 | 1.271–8.177 | 0.015 | 1.429 | 0.435–4.72 | 0.553 |
| Hypertension | 2.219 | 0.82–7.093 | 0.140 | 0.901 | 0.223–3.83 | 0.883 |
| Diabetes mellitus | 1.887 | 0.694–4.86 | 0.196 | | | |
| Ischemic heart disease | 1.31 | 0.189–5.69 | 0.744 | | | |
| Current smoker | 2.625 | 0.532–10.267 | 0.187 | 2.472 | 0.381–13.845 | 0.313 |
| Former smoker | 7 | 1.762–28.091 | 0.005 | 9.608 | 1.827–56.445 | 0.008 |
| ACE-I | 3.013 | 1.16–7.711 | 0.021 | 3.811 | 1.083–14.578 | 0.041 |
| Angiotensin receptor blockers | 0.825 | 0.224–2.445 | 0.746 | | | |
| PaO2/FiO2 | 0.986 | 0.976–0.994 | 0.003 | 0.99 | 0.978–1 | 0.073 |
| Respiratory rate | 1.09 | 1.023–1.169 | 0.010 | 1.07 | 0.997–1.159 | 0.076 |
| CRP | 1.496 | 0.957–2.804 | 0.110 | 1.474 | 0.892–2.673 | 0.119 |
| Lung involvement 20–40% | 0.533 | 0.122–2.195 | 0.381 | | | |
| Lung involvement 40–60% | 1.255 | 0.357–4.699 | 0.724 | | | |
| Lung involvement >60% | 1.517 | 0.427–5.738 | 0.522 | | | |
| Glucocorticoids before TCZ | 0.983 | 0.302–2.741 | 0.975 | | | |

therapy, while increasing $PaO_2/FiO_2$ was positively associated with response to therapy (OR for unit increase = 0.995; 95%CI = 0.98–1, p = 0.0013). No significant difference in response rates was observed between patients treated with intravenous or subcutaneous tocilizumab.

Regarding the composite outcome death/intubation (Table 3), the variables selected for the multivariate analysis were age, clinical status at the beginning of follow-up, hypertension, smoke, use of ACE-I, $PaO_2/FiO_2$, respiratory rate and CRP.

The use of ACE-I and a past history of smoke were significantly related to unfavourable outcome, thus considered as risk factors for death or intubation (OR = 3.81, 95%CI = 1.16–7.71, p = 0.02, and OR = 9.6, 95%CI = 1.76–28.1, p = 0.005, respectively). Moreover, a borderline association was found for respiratory rate as a risk factor for death/intubation (OR for unit increase = 1.07, 95%CI = 1.02–1.17, p = 0.01), while an increase of $PaO_2/FiO_2$ had a trend towards significance as a protective factor for the composite outcome (OR for unit increase = 1, 95%CI = 0.98–0.99, p = 0.003).

Finally, response model achieved an AUC of 0.72 (0.95CI: 0.62–0.82) while death/intubation had an AUC of 0.83 (0.95CI: 0.73–0.93).

Regarding safety analysis one patient had an Acinetobacter baumannii sepsis and another one had oesophageal candidiasis, both in non-responders. No other opportunistic infection was detected.

## Discussion

Our study showed that most patients improved after TCZ therapy (72.9%), while a total of 25 patients died or were intubated during the follow-up.

In our study many clinical, serological and radiological data have been analysed. Nevertheless, none of them was significantly associated with improvement after TCZ therapy. High CRP levels had a borderline association with a negative outcome.

On the contrary, some interesting findings emerged regarding the composite outcome death/intubation, as the use of ACE-I and a past history of smoke were found to be significant risk factors. Moreover, the unit increase of respiratory rate was significantly related to unfavourable outcome, while an increment of PaO2/FiO2 had a trend towards significance as a protective factor for the composite outcome. However, all these associated factors might be the expression of more severe disease instead than negative response to TCZ predictors.

We previously reported that higher CRP levels before starting TCZ and 3 days after the onset of the disease were associated with a reduced therapy response and an increased risk of being intubated or die [17]. Furthermore, CRP levels persisted elevated in non-responders and in patients who were intubated or died [17]. Higher CRP levels suggest a high systemic inflammation. We can speculate that in most severe clinical pictures, where a massive immune activation is sustained by several inflammatory pathways, TCZ might be not sufficient to downregulate the hyperinflammation as it targets a single inflammatory pathway. In fact, even though high levels of IL6 were assessed as a risk factors for severe forms of COVID19, we previously not observe any association with IL-6 levels and response to TCZ therapy [17]. Consistently with this interpretation, the RECOVERY trial showed an improved survival in patients treated with TCZ only in the group treated also with glucocorticoids, whereas no improvement was shown in the group not treated with glucocorticoids [9].

In our study, the use of ACE-I was found to be significant risk factors for death or intubation. Both negative and positive effects of ACE-I on COVID-19 clinical outcomes have been described. However, it remains unclear whether their use affects the clinical course of SARS-CoV2 infection [18].

Notably, tocilizumab therapy showed a good safety profile. Nevertheless, it must be noticed that during the first months of the COVID-19 the awareness regarding secondary opportunistic infection was low and no specific screening for pulmonary aspergillosis was in place.

The current study has many limitations, but also some strengths. This was a retrospective observational cohort study conducted at a two-hospital healthcare organisation following the same internal protocol for the management of COVID19 pneumonia. The limitation could be the sample size and its retrospective nature. However, it represents one of the largest series of TCZ-treated patients and all patients were homogeneously followed-up using a common standardized protocol.

Another strength is that we analysed many clinical, serological and radiological data (Tables 2 and 3) in order to evaluate a great number of variables. Finally, not having an untreated control group, we cannot distinguish if the identified biomarkers are predictors of COVID-19 prognosis even independently of TCZ treatment.

In conclusion, even evaluating a large number of clinical variables, our study did not find new predictors of outcome in COVID-19 pneumonia patients treated with TCZ. It must be noted that 35.2% of our patients did not receive glucocorticoids-treatment and that the combination of glucocorticoids and TCZ appears to act synergistically and to be particularly beneficial in patients with severe COVID-19 and systemic inflammation.

Further studies are needed to investigate the use of this drugs in COVID-19 pneumonia and to better identify clinical phenotypes which could benefit from tocilizumab therapy.

## Author Contributions

**Conceptualization:** Giulia Cassone, Giovanni Dolci, Giulia Besutti, Paolo Pavone, Fabio Sampaolesi, Elisabetta Teopompi, Marco Massari, Matteo Fontana, Stefania Croci, Emanuele Alberto Negri, Massimo Costantini, Nicola Facciolongo, Carlo Salvarani.

**Data curation:** Giulia Cassone, Giovanni Dolci, Giulia Besutti, Luca Braglia, Romina Corsini, Fabio Sampaolesi, Valentina Iotti, Elisabetta Teopompi, Matteo Fontana, Giulia Ghidoni, Anaflorina Matei, Stefania Croci, Emanuele Alberto Negri.

**Formal analysis:** Luca Braglia, Romina Corsini.

**Methodology:** Luca Braglia, Romina Corsini, Carlo Salvarani.

**Project administration:** Giulia Cassone.

**Supervision:** Massimo Costantini, Carlo Salvarani.

**Writing – original draft:** Giulia Cassone, Giovanni Dolci, Giulia Besutti, Luca Braglia.

**Writing – review & editing:** Giulia Cassone, Giovanni Dolci, Giulia Besutti, Luca Braglia, Paolo Pavone, Romina Corsini, Fabio Sampaolesi, Valentina Iotti, Elisabetta Teopompi, Marco Massari, Matteo Fontana, Giulia Ghidoni, Anaflorina Matei, Stefania Croci, Emanuele Alberto Negri, Massimo Costantini, Nicola Facciolongo, Carlo Salvarani.

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
