## [Decision Letter · Decision Letter 0]

8 Jun 2021

PONE-D-21-11985

Clinical outcomes predictive factors in patients with COVID-19 treated with tocilizumab: a monocentric retrospective analysis.

PLOS ONE

Dear Dr. Dolci,

Thank you for submitting your manuscript to PLOS ONE. After careful consideration, we feel that it has merit but does not fully meet PLOS ONE’s publication criteria as it currently stands. Therefore, we invite you to submit a revised version of the manuscript that addresses the points raised during the review process.

We look forward to receiving your revised manuscript.

Kind regards,

Tai-Heng Chen, M.D.

Academic Editor

PLOS ONE

Journal Requirements:

2. Please provide additional details regarding participant consent. In the Methods section, please state why it was not possible to obtain written consent, how verbal consent was recorded and whether the ethics committee approved this consent procedure. If your study included minors, state whether you obtained consent from parents or guardians.

3.We note that you have indicated that data from this study are available upon request. PLOS only allows data to be available upon request if there are legal or ethical restrictions on sharing data publicly. For information on unacceptable data access restrictions, please see http://journals.plos.org/plosone/s/data-availability#loc-unacceptable-data-access-restrictions.

Reviewers' comments:

Reviewer's Responses to Questions

**Comments to the Author**

1. Is the manuscript technically sound, and do the data support the conclusions?

Reviewer #1: Partly

Reviewer #2: Partly

Reviewer #3: Partly

2. Has the statistical analysis been performed appropriately and rigorously? 

Reviewer #1: Yes

Reviewer #2: Yes

Reviewer #3: No

3. Have the authors made all data underlying the findings in their manuscript fully available?

Reviewer #1: Yes

Reviewer #2: No

Reviewer #3: Yes

4. Is the manuscript presented in an intelligible fashion and written in standard English?

Reviewer #1: Yes

Reviewer #2: Yes

Reviewer #3: Yes

5. Review Comments to the Author

Reviewer #1: Dr Dolci, and colleagues addressed a nice study, “Clinical outcomes predictive factors in patients with COVID-19 treated with tocilizumab: a monocentric retrospective analysis”. However, there are many concerns that must be address.

Materials, methods, and Results

Major concerns:

1. The glucocorticoids (GCs) along with or after TCZ should be included in the uni and multi models at table 1 and 2 analysis, once 42 and 50% of each group were on GCs therapy

2. The median follow-up of 12 days (range 8-19 days) is too short to evaluate the outcomes, the authors should extent at least until 28 days.

3. The authors decided to include TCZ in subcutaneous formulation, please provide different analysis for each one, once the PK/PD could be different, and based on result of this analysis, please add in discussion section an explanation supporting or not its use.

Minor concerns:

1. The authors should include all abbreviation with meanings at the first time “CT scans performed at ER (???)” line 145, page 6

Reviewer #2: This is a single center retrospective review evaluating outcomes and predictor factors for the use of tocilizumab and COVID-19 infections. The study had a fairly large population of n=144. As this is a retrospective review, the conclusions and causality are difficult to infer. It is an interesting study as it proceeded the use of remdesivir.

I did have some comments/suggestions for the authors:

Other comorbidities associated with severe COVID were not included: pulmonary disease, stroke, kidney disease, and liver disease. What was the outcome of the non-responders who did not die? Were any of the individuals immunosuppressed at baseline? What was the cause of death for non-responders? Any further delineation on how sick patients were on admission (i.e. SOFA score) would be useful to understand the patient population in the study.

Any report of co-infections or opportunistic infections in the non-responders?

I would like further delineation on the use of SQ versus IV tocilizumab. Was there any difference in these groups and outcomes and deaths?

Additionally, including the median time of initiation of symptoms and/or the median time of admission to administration tocilizumab therapy would be helpful in the responder versus non responder groups.

I was also confused there is a variable in Table 3 of NIV or mechanical ventilation at time of tocilizumab therapy but I thought non-responders were identified as requiring mechanical ventilation or death.

Table 1: Would consider using decimal points instead of commas for p-value column. Consider adding a column for responders as this should be the direct comparison to non-responders. It is less useful to compare non-responder to total population.

Reviewer #3: Although the paper does highlight the paucity of information available on the utility of tocilizumab, unfortunately there are several flaws to the study.

1. The wording of the title is confusing. It would be preferable to read “Predictive factors of clinical outcomes in patients with COVID-19 …” rather than Clinical outcomes predictive factors in patients with COVID-19…

2. Multiple times throughout the paper, the writers use the term “heavy” – line 37, 64, 288 etc. It is unclear what is meant by this term and I suspect it may be a mistranslation.

3. I am concerned about the difference in administration of the tocilizumab – some patients received the drug subcutaneously and some intravenously. I believe a subanalysis demonstrating no difference between the two formulations would be prudent given that how the drug is administered may impact outcomes. Additionally, there does not seem to be any subanalysis to determine if there are differences that could be attributed to patients who received other treatments, including hydroxychloroquine, lopinavir/ritonavir or darunavir/cobicistat

4. Line 145 has “???”. I suspect something else was meant to be written here.

5. The six-category ordinal scale used as an outcome measure was not described

6. Overall, this is a small study that does not include appropriately defined outcome measures nor does it explore potential biases appropriately. The conclusions are not substantially different from the larger trials that the authors themselves cite, including REMAP-CAP and RECOVERY.

6. PLOS authors have the option to publish the peer review history of their article (what does this mean?). If published, this will include your full peer review and any attached files.

Reviewer #1: No

Reviewer #2: No

Reviewer #3: No

---

## [Author Response · Author response to Decision Letter 0]

31 Oct 2021

MANUSCRIPT ID: PONE-D-21-11985R1

23rd July 2021

Dear Editor, 

We are very grateful for your constructive comments and suggestions to our paper entitled “Clinical outcomes predictive factors in patients with COVID-19 treated with tocilizumab: a monocentric retrospective analysis.”.

We here provide a point-by-point reply to the comments and we have incorporated the related changes in the manuscript. The page and lines reported refer to the manuscript with changes track. We thank for the thoughtful insights which helped to significantly improve the manuscript. 

Reviewer #1: Dr Dolci, and colleagues addressed a nice study, “Clinical outcomes predictive factors in patients with COVID-19 treated with tocilizumab: a monocentric retrospective analysis”. However, there are many concerns that must be address.

Materials, methods, and Results

Major concerns:

1. The glucocorticoids (GCs) along with or after TCZ should be included in the uni and multi models at table 1 and 2 analysis, once 42 and 50% of each group were on GCs therapy.

Authors’ reply: we are grateful for this comment. We have performed this analysis and added it to table 1. No associations with the outcomes were found.

2. The median follow-up of 12 days (range 8-19 days) is too short to evaluate the outcomes, the authors should extent at least until 28 days. 

Authors’ reply: we are grateful for this comment and we extended the follow-up to the last data available for each patient in our electronic system. The new median follow-up is 318 (IQR 51-428.5), but no variation in the primary outcome was witnessed. We updated the data in the manuscript (table 1 and page 18, line 203).

3. The authors decided to include TCZ in subcutaneous formulation, please provide different analysis for each one, once the PK/PD could be different, and based on result of this analysis, please add in discussion section an explanation supporting or not its use. 

Authors’ reply: we are grateful for these comments and we explored this approach previously but we did not include it in the paper as splitting the analysis by type of administration would be detrimental for sample size/power of each analysis. However, we formally added administration type in the variable selection procedure (last line of table 2) and no differences in response/death-intubation were associated to it. We also added this sentence at lines 249-250: “No significant difference in response rates was observed between patients treated with intravenous or subcutaneous tocilizumab.”

Minor concerns:

1. The authors should include all abbreviation with meanings at the first time “CT scans performed at ER (???)” line 145, page 6. 

Authors’ reply: we modified the paper accordingly.

Reviewer #2: This is a single center retrospective review evaluating outcomes and predictor factors for the use of tocilizumab and COVID-19 infections. The study had a fairly large population of n=144. As this is a retrospective review, the conclusions and causality are difficult to infer. It is an interesting study as it proceeded the use of remdesivir. 

I did have some comments/suggestions for the authors:

Other comorbidities associated with severe COVID were not included: pulmonary disease, stroke, kidney disease, and liver disease. 

Authors’ reply: regarding pulmonary diseases COPD was included in the first version of the manuscript (Table 1). Regarding the other conditions, we added “Three patients had history of stroke or transient ischemic attack. Five patients had chronic kidney disease and one was a kidney transplant recipient. One patient had cirrhosis.” (page 10, line 203-205). 

What was the outcome of the non-responders who did not die?

Authors’ reply: non-responders who did not die underwent a subsequent clinical improvement. To clarify the responder/non responder definition we added this sentence “Patients who initially declined and afterward improved on the six-category ordinal scale were defined as non-responders.” at pag. 7, line 161-162.

Were any of the individuals immunosuppressed at baseline?

Authors’ reply: at page 18, line 205-207, we added “Only two patients were on immunosuppressive therapy at baseline, one for ulcerative rectum-colitis and one for polymyalgia rheumatica. One patient had HIV infection.”

What was the cause of death for non-responders?

Authors’ reply: the cause of death for non responders was respiratory failure or intensive care-related complications.

Any further delineation on how sick patients were on admission (i.e. SOFA score) would be useful to understand the patient population in the study. 

Authors’ reply: we added the sentence “Median Charlson’s score was 2 (range 0-8, IQR 2-3).” at pag. 7-8, lines 182-183. 

Any report of co-infections or opportunistic infections in the non-responders? 

Authors’ reply: we added this sentence in the results section, page 23, lines 266-267: “Regarding safety analysis one patient had an Acinetobacter baumanii sepsis and another one had oesophageal candidiasis, both in non-responders. No other opportunistic infection was detected.”. We have discussed this result at page 24, lines 300-302 as follow: “Notably, tocilizumab therapy showed a good safety profile. Nevertheless, it must be noticed that during the first months of the COVID-19 the awareness regarding secondary opportunistic infection was low and no specific screening for pulmonary aspergillosis was in place.”

I would like further delineation on the use of SQ versus IV tocilizumab. Was there any difference in these groups and outcomes and deaths? 

Authors’ reply: we are grateful for this comment. As previously answered to Reviewer #1 we explored this approach previously but we did not include it in the paper as splitting the analysis by type of administration would be detrimental for sample size/power of each analysis. However, we formally added administration type in the variable selection procedure (last line of table 2) and no differences in response/death-intubation were associated to it. We also added this sentence at page 22 lines 249-250: “No significant difference in response rates was observed between patients treated with intravenous or subcutaneous tocilizumab.”

Additionally, including the median time of initiation of symptoms and/or the median time of admission to administration tocilizumab therapy would be helpful in the responder versus non responder groups.

Authors’ reply: we performed the suggested analysis, a slightly lower time from symptom onset to admission/first treatment was recorded for death/int patients (6 days in median) compared to the others (7, p = 0.038), likely due to more severe symptoms and quickier admission/treatment. No difference was found. We included the analysis in table one.

I was also confused there is a variable in Table 3 of NIV or mechanical ventilation at time of tocilizumab therapy but I thought non-responders were identified as requiring mechanical ventilation or death. 

Authors’ reply: non responders were defined as “non-response to treatment was defined as at least one-point worsening of the clinical status after TCZ therapy.” (page 7, lines 161-162).

Table 1: Would consider using decimal points instead of commas for p-value column. Consider adding a column for responders as this should be the direct comparison to non-responders. It is less useful to compare non-responder to total population. 

Authors’ reply: thank you for this comment, we modified table 1 accordingly.

Reviewer #3: Although the paper does highlight the paucity of information available on the utility of tocilizumab, unfortunately there are several flaws to the study. 

1. The wording of the title is confusing. It would be preferable to read “Predictive factors of clinical outcomes in patients with COVID-19 …” rather than Clinical outcomes predictive factors in patients with COVID-19… 

Authors’ reply: we are grateful for this suggestion, we modified the title accordingly: “Predictive factors of clinical outcomes in patients with COVID-19 treated with tocilizumab: a monocentric retrospective analysis.”. 

2. Multiple times throughout the paper, the writers use the term “heavy” – line 37, 64, 288 etc. It is unclear what is meant by this term and I suspect it may be a mistranslation. 

Authors’ reply: we are grateful for this suggestion, we modified the paper accordingly.

3. I am concerned about the difference in administration of the tocilizumab – some patients received the drug subcutaneously and some intravenously. I believe a subanalysis demonstrating no difference between the two formulations would be prudent given that how the drug is administered may impact outcomes. Additionally, there does not seem to be any subanalysis to determine if there are differences that could be attributed to patients who received other treatments, including hydroxychloroquine, lopinavir/ritonavir or darunavir/cobicistat Braglia, poi Giovanni

Authors’ reply: we are grateful for this comment. We have performed an analysis considering different ways of tocilizumab administration and added it to table 1. No associations with the outcomes were found. 

4. Line 145 has “???”. I suspect something else was meant to be written here. 

Authors’ reply: we are grateful for this comment and we apologize for the typo. We have changed the paper accordingly.

5. The six-category ordinal scale used as an outcome measure was not described. 

Authors’ reply: the six-category ordinal scale used was described in the original manuscript and you can find it at pages 6-7, lines 154-157 of the revised version.

6. Overall, this is a small study that does not include appropriately defined outcome measures nor does it explore potential biases appropriately. The conclusions are not substantially different from the larger trials that the authors themselves cite, including REMAP-CAP and RECOVERY. 

Authors’ reply: whereas we do agree that the conclusions are not substantially different from larger trials, we think that outcome measures were appropriately defined and potential biases explored in the context of the limitations that we have reported in the discussion. Furthermore, even though larger multicenter cohort studies and trials have been published, we describe one of the largest monocentric cohort of patients with COVID-19 treated with tocilizumab. 

We would like to express our great appreciation to you and the reviewers for the comments on our paper. If you have any further queries, please do not hesitate to contact us. 

Kind regards, 

Giovanni Dolci

---

## [Decision Letter · Decision Letter 1]

10 Jan 2022

Predictive factors of clinical outcomes in patients with COVID-19 treated with tocilizumab: a monocentric retrospective analysis.

PONE-D-21-11985R1

Dear Dr. Dolci,

We’re pleased to inform you that your manuscript has been judged scientifically suitable for publication and will be formally accepted for publication once it meets all outstanding technical requirements.

Kind regards,

Tai-Heng Chen, M.D.

Academic Editor

PLOS ONE

Reviewers' comments:

Reviewer's Responses to Questions

**Comments to the Author**

1. If the authors have adequately addressed your comments raised in a previous round of review and you feel that this manuscript is now acceptable for publication, you may indicate that here to bypass the “Comments to the Author” section, enter your conflict of interest statement in the “Confidential to Editor” section, and submit your "Accept" recommendation.

Reviewer #1: All comments have been addressed

2. Is the manuscript technically sound, and do the data support the conclusions?

Reviewer #1: Yes

3. Has the statistical analysis been performed appropriately and rigorously? 

Reviewer #1: Yes

4. Have the authors made all data underlying the findings in their manuscript fully available?

Reviewer #1: Yes

5. Is the manuscript presented in an intelligible fashion and written in standard English?

Reviewer #1: Yes

6. Review Comments to the Author

Reviewer #1: The authors addressed all answers to the comments . I don’t need of further information and the manuscript is fine in the current format.

7. PLOS authors have the option to publish the peer review history of their article (what does this mean?). If published, this will include your full peer review and any attached files.

Reviewer #1: No

---

## [Editor Report · Acceptance letter]

17 Jan 2022

PONE-D-21-11985R1 

Predictive factors of clinical outcomes in patients with COVID-19 treated with tocilizumab: a monocentric retrospective analysis. 

Dear Dr. Dolci:

I'm pleased to inform you that your manuscript has been deemed suitable for publication in PLOS ONE. Congratulations! Your manuscript is now with our production department. 

Kind regards, 

on behalf of

Dr. Tai-Heng Chen 

Academic Editor

PLOS ONE